# Agreement on emotion labels' frequency in eight Spanish linguistic areas

**Ana R. Delgado**[1]*, **Gerardo Prieto**[1], **Debora I. Burin**[2]

**1** Facultad de Psicología, Universidad de Salamanca, Salamanca, Spain, **2** Facultad de Psicología, Universidad de Buenos Aires-CONICET, Buenos Aires, Argentina

* adelgado@usal.es

**Data Availability Statement:** All relevant data are within the paper (Table 1 and Table 3).

**Funding:** The author(s) received no specific funding for this work.

## Abstract

Various traditions have investigated the relationship between emotion and language. For the basic emotions view, emotional prototypes are lexically sedimented in language, evidenced in cultural convergence in emotional recognition and expression tasks. For constructionist theories, conceptual knowledge supported by language is at the core of emotions. Understanding emotion words is embedded in various interrelated constructs such as emotional intelligence, emotion knowledge or emotion differentiation, and is related to, but different from, general vocabulary. A clear advantage of Emotion Vocabulary over most emotion-related constructs is that it can be measured objectively. In two successive corpus-based studies, we tested the predictions of concordance and absolute agreement on the frequency of use of a total of 100 Spanish emotion labels in the eight main Spanish-speaking areas: Spain, Mexico-Central America, River Plate, Continental Caribbean, Andean, Antilles, Chilean, and the United States. In both studies, the intraclass correlation coefficient was statistically different from the null and very large, over .95, as was the Kendall's concordance coefficient, indicating broad consensus among the Spanish linguistic areas. From an applied perspective, our results provide supporting evidence for the similarity in frequency, and therefore cross-cultural generalizability regarding familiarity of the 100 emotion labels as item stems or as experimental stimuli without going through a process of additional adaptation. On a broader scope, these results add evidence on the role of language for emotion theories. In this regard, countries and regions compared here share the same Spanish language, but differ in several aspects in history, culture, and socio-economic structure.

## Introduction

The traditional view of emotions posits that they are basic, universal, phylogenetically shaped processes that are engrained in human biological functioning, and thus organize cognitive, experiential, and behavioural reactions to changes in the environment [1]. Emotions encompass physiology, actions, facial, vocal, and postural expression, and cognitive processes, and have both a rapid response and a social interaction function. Emotional episodes and experiences would conform to universal prototypes, with cultural variations but within a general

**Competing interests:** The authors have declared that no competing interests exist.

categorical similarity [2]. Emotional prototypes would be lexically sedimented in language, evidenced in cultural convergence in emotional recognition and expression tasks employing emotional words alone, or in short verbal statements [3, 4]. Although these tasks have been criticized because languages vary in words that refer to specific emotions, and some supposed basic emotions do not have a specific word in some cultures [5], studies on emotional recognition and labelling often find agreement.

A different view of emotions, the constructionist perspective, proposes that conceptual knowledge supported by language is at the core of emotions [5–7]. The summary representation of any emotion category is an abstraction, not a denomination of a natural object such as a body or brain state. Interoceptive sensations, experienced as lower dimensional feelings of affect (valence and arousal), are assumed to be in continuous interpretation, along with other sensory and motor inputs and outputs, by a predictive brain that implements conceptual categories in its internal model to give them meaning [7]. The brain uses emotion concepts to categorize sensations, and to dynamically construct various instances of emotion in specific situations. Socio-cultural mechanisms, especially language, are responsible for organizing and differentiating emotional experiences [6–9]. Language is the "glue" that helps to link bodily states, perceptions of muscular movements in the face and body, and other sensory and motor experiences as instances of a particular emotion concept. For example, emotion words and their associated semantic knowledge have been shown to determine how facial configurations are predicted, encoded, and remembered as emotional expressions [10].

Emotion vocabulary grows in childhood and adolescence, with individual and group differences [11–14]. Understanding of the emotion vocabulary is embedded in various interrelated constructs such as emotional intelligence [15, 16], emotion knowledge [17], or emotion differentiation [18]. Understanding emotion words is an integral part of emotional intelligence, and is related to, but different from, general vocabulary [15]. Being able to distinguish between affective experiences, and to label negative ones, is associated with several indices of mental health in adulthood, and more adaptive emotion regulation [18]. For instance, in a study with the experiential sampling technique, in which participants reported several times a day their emotional experience, patients with social anxiety disorder had less differentiated negative emotions compared to controls, controlling for intensity and comorbidity, suggesting an association between the anxiety disorder and understanding emotions at a given moment in daily life [19]. For people who experience a traumatic or negative episode, talking and writing about their emotion acts as a buffer for mental health [20].

The relevance of emotion vocabulary in emotion theory and in individual differences has led to various measurement instruments (e.g. for adults, vignette-based: *MSCEIT* [16]; *STEU* [15]; or word definition: *GEMOK-Features* [21]. In Spanish, the Emotion Vocabulary Test (EVT) was recently developed [22, 23]. Each of the 40 multiple-choice items of the EVT is composed of a Spanish emotion label (the item stem) and five response options corresponding to the five broad emotion "families" of happiness, sadness, anger, fear, and disgust.

In principle, the use of these 40 Spanish emotion words (the EVT item stems) as psychometric or experimental stimuli in other Spanish-speaking zones would require an adaptation procedure. Under the unitary umbrella of construct validity, content validation strategies are appropriate when the boundaries of a domain can be described [24], as is the case with emotion vocabulary. It is common to have experts to adapt test content to other languages or cultures. Here, we propose a less subjective procedure based on a corpus approach. Linguistic corpora analyses have been employed for studying language and cultural comparisons [25] and have gained traction in this century due to the vast linguistic information online and the availability of computerized and big data analytic tools [26, 27]. For example, [28] calculated

the co-occurrence in a corpus of unselected text from USENET discussion groups, of emotion words taken from basic emotion models.

In the present case, we have focused on the lexical level, and the CORPES XXI corpus [29]. Spanish is the second most spoken mother tongue, with 460 million native speakers in 31 countries [30]. There are eight main Spanish-speaking areas: Spain, Mexico-Central America, River Plate, Continental Caribbean, Andean, Antilles, Chilean, and USA; they all are represented into the Spanish Corpus of the Royal Academy, CORPES XXI with about 300 million forms from oral (10%) and written text (40% from books, 40% from periodicals, 7.5% internet material, and 2.5% miscellaneous). Of the texts, 30% are from Spain [29]. Note that the absolute frequency of a word in one linguistic area should not be compared with the absolute frequency of that word in another area because they are not equally represented in the corpus. This is why CORPES XXI also offers the possibility of obtaining normalized frequencies per million words, i.e., relative frequencies in each area multiplied by one million (*fpmw*).

Initial steps in corpora analyses generate frequency lists, to map out and compare word frequency across either an entire corpus or across particular sub-sets (sub-corpora). Although this would not constitute a deep semantic analysis, it is a first step in comparing the lexical structure, and possible lexical (and cultural) differences. A positive answer to the question "Is there consensus in frequency for the 40 Spanish emotion labels (EVT stems) comparing the eight Spanish speaking linguistic areas?" would provide supporting evidence for the use of the EVT item stems as psychometric or experimental stimuli in any Spanish-speaking area before additional adaptation. In other research settings, it could provide a set of emotion labels with similar frequency across Spanish speaking countries. Frequency is one of the main factors affecting several experimental psycholinguistic outcomes, such as lexical decision, word naming, language comprehension, and memory recall and recognition [31, 32].

On a broader scope, it would add evidence regarding the role of language for emotion theories. In this regard, countries and regions compared here share the same Spanish language, but differ in several aspects in history, culture, and socio-economic structure. Although frequency does not reflect semantic meaning, it is one of the basic dimensions of a lexicon, indicating ease of access; its effects reflect in part semantic activation, given that lexical access is mediated by the number of contexts in which a word tends to occur rather than pure repetition of occurrence [32].

A second study, if consensus results were replicated, would help to content-validate new items/stimuli as well as to reinforce the conclusions of the first study.

Thus, two successive corpus-based studies (CORPES XXI [29]) were carried out to test the predictions of concordance and absolute agreement on the frequency of use of a total of 100 Spanish emotion words –40 emotion labels from the EV test (Study 1) and 60 new emotion labels (Study 2)–in the eight main Spanish-speaking areas (Spain, Mexico-Central America, River Plate, Continental Caribbean, Andean, Antilles, Chilean, and the United States).

## Materials and methods

The geographical distribution of the forms in CORPES XXI v. 0.91 for the eight main areas was: Spain (32%), Mexico-Central America (19%), River Plate (14%), Continental Caribbean (12%), Andean (8%), Antilles (7%), Chilean (6%), USA (1%). We did not take into account areas whose representation was under 0.5% (Guinea and the Philippines).

A simple way of testing concordance among areas ("judges") regarding the *order* of emotion words ("objects") is by using the Kendall Coefficient of Concordance (W): Let us think of the emotion words as "objects" and then think of the various areas as the "judges" that rank them. Only the ranks are now less subjective, not coming from expert judgement but from word frequency. The W statistic does not require the assumption of quantitative scaling. Considering *fpmw* as quantitative,

we can also assess absolute agreement by means of an Intra-class Correlation Coefficient (ICC), a measure of the proportion of variance that can be attributed to the measurement objects [33].

There are various ICC kinds depending on the answers to three questions: Do the same "judges" score every "object"? Are "judges" a sample or a population? Is reliability of a single "judge" or of their average? For our data (i.e., "judges" are the 8 main Spanish linguistic areas, "objects" are the 40 words (Study 1) or 60 words (Study 2), each "receiving" a *fpmw*), ICC kinds would correspond to the following models:

- One-*way random effects*: each word *fpmw* is given in different areas that are sampled from a larger pool of potential areas that are treated as random effects.

- *Two-way random effects*: all word *fpmw* are calculated in all areas; both factors–areas and words–are random effects. It is a consistency coefficient (C-type ICC).

- *Two-way mixed effects*: areas are considered as fixed effects but words are treated as random effects. It is an absolute agreement coefficient (A-type ICC).

They are called ICC(1), ICC(C,1), and ICC(A,1) respectively when the unit of analysis is the individual, and ICC(k), ICC(C,k), and ICC(A,k) when it is an average (of k "judges"). Because our objective was to test the hypothesis of consensus regarding the frequency of use of the emotion labels among the 8 linguistic areas, finding ordinal concordance would constitute soft evidence. A large-sized absolute agreement ICC value, over .90, would be considered as strong evidence to corroborate our hypothesis.

The Kendall coefficient of concordance (W), and ICC(A,8) for absolute agreement (Spanish linguistic areas are a fixed-effect factor) were calculated by means of the R package [34] "irr" [35] on the RStudio environment [36]. In addition to these two statistics, and just for comparison purposes, we report results for the remaining ICC *two-way* models.

## Study 1

### Materials and procedure

CORPES XXI *normalized frequencies per million* for the 40 Spanish emotion labels (the stems of the 40 multiple-choice items of the EVT) in each of the 8 main linguistic areas were retrieved on December the 5th, 2018. They can be seen from Table 1, where both words and areas are in alphabetical order.

### Results

The Kendall coefficient of concordance was statistically different from the null, *W* = .960, *Chi-squared(39)* = 300, *p* < .001, and very large-sized, as was the *ICC (A,8)* = 0.995 [*F*(39,231) = 226, *p* < .001, *95% CI*: 0.993 < ICC < 0.997] indicating absolute agreement, i.e., broad consensus among the eight Spanish linguistic areas.

Different assumptions regarding the various ICC kinds would not change this conclusion, as can be seen from Table 2. The 95% confidence intervals make clear that they all are well over the .90 that we consider would show strong evidence of consensus among areas for the frequency of use of the 40 EVT stem words.

## Study 2

### Materials and procedure

A list of another 60 Spanish emotion labels was made by looking for synonyms of the EVT stems as well as words from the Spanish semantic field of the empirically-derived English

**Table 1. Forty emotion label *fpmw* by linguistic area.**

| Word | Andean | Antill. | Carib. | Chile | Mexico | River. | Spain | USA |
|------|--------|---------|--------|-------|--------|--------|-------|-----|
| aflicción | 2.04 | 1.17 | 1.31 | 1.49 | 2.09 | 1.38 | 1.35 | 0.32 |
| amargura | 9.83 | 6.78 | 8.02 | 4.89 | 7.27 | 5.70 | 8.15 | 4.50 |
| angustia | 25.15 | 20.78 | 31.30 | 32.92 | 29.91 | 33.40 | 24.85 | 19.96 |
| aversión | 2.74 | 1.17 | 3.00 | 1.95 | 2.89 | 1.57 | 3.20 | 0.96 |
| cólera (fem.) | 4.89 | 4.86 | 3.07 | 1.56 | 4.76 | 2.49 | 3.76 | 1.28 |
| contento | 0.09 | 0.10 | 0.37 | 0.19 | 0.19 | 0.16 | 0.25 | 0.00 |
| desagrado | 2.09 | 1.60 | 2.19 | 2.73 | 2.15 | 2.27 | 2.53 | 1.28 |
| desaire | 0.69 | 0.96 | 0.72 | 1.17 | 0.77 | 0.74 | 0.64 | 0.00 |
| desasosiego | 2.99 | 6.57 | 3.97 | 2.54 | 3.96 | 3.07 | 5.60 | 1.28 |
| desconsuelo | 1.74 | 2.19 | 2.66 | 1.63 | 2.39 | 2.32 | 2.04 | 0.96 |
| desdén | 4.59 | 5.02 | 5.29 | 3.65 | 4.74 | 3.60 | 4.81 | 1.28 |
| desolación | 4.19 | 3.58 | 5.51 | 8.99 | 4.80 | 4.29 | 6.02 | 2.89 |
| desprecio | 12.27 | 13.35 | 14.06 | 12.51 | 14.91 | 15.67 | 16.75 | 5.15 |
| dicha | 5.34 | 9.19 | 9.99 | 4.17 | 8.25 | 5.92 | 4.37 | 5.47 |
| duelo | 23.50 | 22.28 | 27.98 | 35.01 | 40.08 | 23.15 | 23.65 | 36.70 |
| entusiasmo | 31.94 | 35.00 | 27.51 | 33.25 | 28.61 | 32.93 | 29.64 | 28.97 |
| espanto | 7.28 | 5.50 | 7.33 | 7.82 | 6.34 | 9.97 | 5.86 | 1.28 |
| exasperación | 0.44 | 0.32 | 1.00 | 0.26 | 0.75 | 0.99 | 0.90 | 0.32 |
| exultación | 0.09 | 0.05 | 0.12 | 0.06 | 0.09 | 0.02 | 0.03 | 0.00 |
| felicidad | 39.87 | 42.48 | 51.35 | 34.62 | 44.03 | 47.52 | 44.14 | 35.73 |
| grima | 0.04 | 0.80 | 0.59 | 0.00 | 0.19 | 0.02 | 1.02 | 0.32 |
| indignación | 8.93 | 8.01 | 8.49 | 7.23 | 7.13 | 8.97 | 10.85 | 8.37 |
| inquietud | 13.87 | 12.13 | 14.32 | 21.71 | 14.33 | 16.61 | 18.10 | 11.26 |
| irritación | 3.59 | 4.11 | 3.41 | 2.99 | 4.12 | 4.37 | 5.57 | 3.54 |
| júbilo | 5.24 | 7.85 | 4.23 | 2.47 | 16.49 | 2.76 | 3.35 | 4.82 |
| melancolía | 8.18 | 7.16 | 7.64 | 7.69 | 10.78 | 10.08 | 12.42 | 2.89 |
| pánico | 16.91 | 12.39 | 19.23 | 20.40 | 15.65 | 16.84 | 17.97 | 17.70 |
| pena | 81.20 | 83.04 | 85.10 | 93.95 | 82.44 | 75.44 | 90.14 | 78.55 |
| pesadumbre | 0.94 | 1.60 | 2.06 | 1.17 | 2.57 | 1.52 | 2.40 | 0.00 |
| rabia | 27.10 | 21.80 | 43.52 | 40.29 | 24.09 | 33.62 | 25.26 | 13.52 |
| regocijo | 2.89 | 5.55 | 3.63 | 1.95 | 3.29 | 2.07 | 2.40 | 1.60 |
| rencor | 9.38 | 6.94 | 8.67 | 5.41 | 11.34 | 8.44 | 9.72 | 3.86 |
| repugnancia | 2.29 | 2.03 | 1.59 | 1.49 | 2.47 | 2.18 | 2.99 | 0.96 |
| repulsión | 1.64 | 0.85 | 2.22 | 1.04 | 2.23 | 1.57 | 1.59 | 0.96 |
| resentimiento | 6.03 | 5.18 | 7.64 | 5.80 | 6.24 | 6.17 | 5.19 | 2.89 |
| satisfacción | 28.79 | 49.27 | 33.81 | 37.23 | 34.90 | 27.30 | 37.34 | 31.22 |
| sobresalto | 3.04 | 3.90 | 3.63 | 2.67 | 2.87 | 2.52 | 3.33 | 0.64 |
| susto | 11.12 | 11.22 | 14.22 | 10.88 | 11.86 | 9.99 | 12.73 | 7.08 |
| temor | 50.85 | 48.25 | 48.69 | 51.77 | 51.83 | 45.45 | 33.99 | 51.19 |

**Table 2. Intra-class correlation coefficient two-way models (40 Words, 8 Areas).**

| Case | ICC | 95% CI |
|------|-----|--------|
| ICC(C,1) | .966 | .948-.979 |
| ICC(A,1) | .963 | .943-.978 |
| ICC(C,8) | .996 | .993-.997 |
| ICC(A,8) | .995 | .993-.997 |

**Table 3. Sixty emotion label *fpmw* by linguistic area.**

| Word | Andean | Antill. | Carib. | Chile | Mexico | River. | Spain | USA |
|---|---|---|---|---|---|---|---|---|
| aburrimiento | 6.41 | 4.70 | 6.16 | 6.40 | 6.01 | 6.58 | 8.46 | 3.39 |
| admiración | 16.27 | 17.48 | 17.39 | 14.18 | 18.17 | 16.00 | 18.90 | 18.65 |
| adoración | 2.60 | 2.90 | 3.05 | 1.60 | 2.95 | 2.42 | 3.04 | 2.26 |
| alegría | 59.13 | 67.94 | 59.89 | 49.79 | 52.67 | 58.51 | 47.87 | 55.67 |
| alivio | 19.01 | 18.63 | 18.22 | 17.27 | 16.87 | 20.44 | 17.15 | 37.86 |
| anhelo | 8.97 | 8.76 | 11.86 | 13.35 | 11.83 | 10.55 | 7.85 | 4.80 |
| ansiedad | 27.15 | 24.55 | 26.77 | 32.46 | 24.79 | 29.83 | 31.23 | 39.84 |
| antipatía | 1.62 | 1.45 | 1.75 | 1.24 | 1.24 | 1.39 | 1.68 | 0.28 |
| añoranza | 3.57 | 4.35 | 2.87 | 2.13 | 2.51 | 2.16 | 3.45 | 0.84 |
| aprecio | 4.41 | 7.11 | 5.01 | 3.79 | 5.69 | 3.17 | 4.81 | 4.52 |
| apuro | 10.46 | 6.46 | 4.98 | 9.73 | 6.15 | 12.72 | 7.99 | 7.34 |
| arrobamiento | 0.65 | 0.35 | 0.51 | 0.29 | 0.22 | 0.49 | 0.26 | 0.00 |
| asco | 13.01 | 9.21 | 10.68 | 16.20 | 14.11 | 13.98 | 17.06 | 3.10 |
| asombro | 15.01 | 19.19 | 17.53 | 14.18 | 18.65 | 20.00 | 15.13 | 10.17 |
| benevolencia | 1.11 | 1.15 | 1.52 | 0.89 | 1.54 | 1.31 | 2.02 | 1.13 |
| bochorno | 1.72 | 1.80 | 1.92 | 2.01 | 2.64 | 2.03 | 2.67 | 1.97 |
| calma | 31.84 | 24.35 | 27.64 | 37.15 | 28.93 | 25.39 | 28.69 | 25.99 |
| cariño | 39.37 | 30.81 | 24.15 | 45.70 | 30.87 | 22.76 | 37.75 | 32.49 |
| celos | 10.73 | 10.12 | 9.04 | 8.78 | 8.60 | 10.35 | 9.23 | 8.47 |
| comodidad | 12.78 | 11.17 | 17.16 | 12.52 | 11.57 | 14.24 | 12.89 | 23.73 |
| compasión | 7.95 | 11.37 | 10.65 | 11.81 | 9.17 | 7.61 | 10.54 | 13.56 |
| compenetración | 0.65 | 1.40 | 0.71 | 0.41 | 0.60 | 0.69 | 1.03 | 0.56 |
| confianza | 80.71 | 77.41 | 88.80 | 98.34 | 93.15 | 73.97 | 86.04 | 102.86 |
| confusión | 21.80 | 22.44 | 25.08 | 19.58 | 24.28 | 26.43 | 25.48 | 19.21 |
| conmiseración | 0.69 | 0.90 | 1.26 | 0.77 | 1.08 | 0.90 | 1.25 | 0.00 |
| culpabilidad | 4.18 | 6.01 | 3.57 | 2.55 | 4.48 | 2.71 | 6.99 | 12.15 |
| curiosidad | 32.73 | 29.16 | 32.33 | 28.96 | 33.78 | 38.14 | 43.01 | 16.39 |
| depresión | 22.54 | 28.05 | 24.21 | 45.76 | 32.91 | 24.49 | 35.03 | 56.23 |
| desazón | 3.85 | 2.35 | 5.32 | 3.97 | 2.70 | 5.05 | 4.38 | 1.41 |
| deseo | 92.24 | 124.61 | 96.29 | 85.28 | 109.19 | 99.37 | 105.34 | 96.08 |
| diversión | 17.34 | 15.23 | 15.00 | 7.95 | 15.01 | 11.51 | 10.51 | 18.65 |
| embeleso | 0.46 | 0.75 | 0.66 | 0.41 | 0.77 | 0.38 | 0.62 | 0.00 |
| empatía | 4.64 | 2.90 | 5.98 | 5.57 | 4.85 | 4.51 | 6.46 | 2.54 |
| enfado | 1.76 | 0.85 | 1.49 | 0.71 | 2.18 | 0.56 | 8.27 | 0.84 |
| enojo | 4.04 | 4.55 | 2.87 | 8.01 | 10.45 | 11.09 | 1.36 | 14.69 |
| envidia | 13.11 | 13.12 | 14.85 | 11.27 | 13.17 | 11.71 | 16.41 | 6.21 |
| exaltación | 3.53 | 4.91 | 4.75 | 3.50 | 4.19 | 3.76 | 5.28 | 3.10 |
| excitación | 6.09 | 5.76 | 5.67 | 5.46 | 6.58 | 8.56 | 8.53 | 3.67 |
| éxtasis | 7.25 | 6.56 | 7.19 | 10.44 | 6.48 | 5.39 | 7.90 | 5.08 |
| furia | 17.76 | 14.68 | 16.78 | 13.11 | 18.03 | 23.30 | 11.78 | 12.43 |
| gozo | 5.39 | 6.91 | 4.46 | 2.01 | 8.05 | 2.86 | 4.95 | 5.65 |
| horror | 19.89 | 19.64 | 24.41 | 23.32 | 22.20 | 25.19 | 21.21 | 18.65 |
| hostilidad | 4.64 | 8.06 | 7.71 | 4.45 | 4.83 | 5.93 | 6.64 | 3.95 |
| humillación | 5.99 | 7.86 | 8.81 | 8.78 | 8.51 | 9.29 | 9.07 | 4.80 |
| interés | 133.52 | 171.06 | 165.74 | 153.06 | 154.32 | 123.89 | 166.85 | 149.49 |
| miedo | 126.50 | 114.99 | 130.04 | 136.50 | 151.41 | 140.13 | 167.15 | 115.86 |
| nerviosismo | 7.20 | 6.46 | 5.49 | 7.77 | 7.38 | 5.75 | 7.64 | 7.34 |

(*Continued*)

**Table 3.** (Continued)

| Word | Andean | Antill. | Carib. | Chile | Mexico | River. | Spain | USA |
|---|---|---|---|---|---|---|---|---|
| nostalgia | 20.78 | 22.49 | 22.74 | 20.65 | 19.37 | 17.60 | 16.17 | 16.95 |
| odio | 26.40 | 29.56 | 32.53 | 23.68 | 32.05 | 29.19 | 26.94 | 20.91 |
| orgullo | 31.89 | 38.38 | 32.56 | 31.27 | 34.57 | 32.49 | 27.31 | 37.30 |
| relajación | 3.90 | 7.11 | 5.52 | 3.38 | 3.89 | 4.07 | 8.31 | 12.15 |
| respeto | 81.17 | 84.42 | 75.93 | 72.64 | 80.74 | 58.33 | 71.75 | 74.04 |
| serenidad | 9.81 | 6.06 | 9.27 | 6.05 | 8.12 | 7.38 | 10.54 | 4.23 |
| sobrecogimiento | 0.32 | 0.15 | 0.34 | 0.17 | 0.20 | 0.10 | 0.31 | 0.28 |
| sorpresa | 71.32 | 60.92 | 63.86 | 71.75 | 67.98 | 71.47 | 77.65 | 72.91 |
| terror | 24.92 | 21.34 | 30.32 | 28.31 | 29.18 | 30.38 | 23.97 | 17.80 |
| tirria | 0.51 | 0.15 | 0.57 | 0.35 | 0.34 | 0.18 | 0.26 | 0.00 |
| tranquilidad | 29.15 | 23.44 | 34.64 | 32.16 | 30.97 | 28.83 | 28.17 | 27.12 |
| tristeza | 32.63 | 29.11 | 41.29 | 31.87 | 36.63 | 32.39 | 29.36 | 24.86 |
| vergüenza | 31.42 | 26.75 | 26.98 | 36.08 | 25.29 | 34.12 | 31.45 | 16.95 |

emotion labels [37, 38]. Note that, in any language, the number of emotion labels, as opposed to the number of emotion-laden words [39] is very limited. On March the 23rd, 2019, CORPES XXI *normalized frequencies per million* for these 60 Spanish emotion words in each of the 8 main linguistic areas were retrieved (Table 3, in alphabetical order).

## Results

The Kendall coefficient of concordance was statistically different from the null, $W = .963$, *Chi-squared (59) = 454*, $p < .001$, and very large-sized, as was the *ICC (A,8) = 0.996* [$F(59,420) = 285$, $p < .001$, *95% CI*: $0.995 < ICC < 0.998$] indicating absolute agreement, i.e., broad consensus among the eight Spanish linguistic areas.

Different assumptions regarding the various ICC kinds would not change this conclusion, as can be seen from Table 4. As in Study 1, the 95% confidence intervals show that they all are over the .90 that we consider would show strong evidence of consensus among areas for the frequency of use of the 60 emotion labels.

## Discussion

This study employed a linguistic corpus analysis approach to compare the relative frequency of emotion labels in the eight main Spanish-speaking areas (Spain, Mexico-Central America, River Plate, Continental Caribbean, Andean, Antilles, Chilean, and USA) as provided by the CORPES XXI normalized frequencies [29]. We found very high levels of agreement among areas for the frequency of use of the 40 EVT stem words. The reference corpus is the biggest in Spanish, has high representativeness and balance [26], including oral transcriptions, from the

**Table 4. Intra-class correlation coefficient two-way models (60 Words, 8 Areas).**

| Case | ICC | 95% CI |
|---|---|---|
| ICC(C,1) | .973 | .961-.982 |
| ICC(A,1) | .973 | .961-.982 |
| ICC(C,8) | .996 | .995-.998 |
| ICC(A,8) | .996 | .995-.998 |

XXI century [29], so this result is a first step to establish a lexical agreement over these words between these regions, with a big and representative reference corpus.

Our results constitute a first step in validation of the EVT test to be used in any of the Spanish speaking regions, allowing for a further semantic adaptation process. As a measure of vocabulary knowledge, word frequency is one of the main factors for item difficulty [40–42]. These results suggest an agreement in frequency, and thus difficulty, for the five broad emotion "families" of happiness, sadness, anger, fear, and disgust and their associated 40 items presented in the test. However, in multiple-choice formats, semantic similarity between the correct answer and distractors, and distractor word frequency and other properties, are also relevant for item difficulty. As a test the EVT might need finer tuning. Future corpora studies can study lexical associations between item words, within and between Spanish speaking regions (e.g. [28]) or compare those results with different participant samples.

Frequency is one of the main factors affecting several psycholinguistic and memory tasks [31, 32]. Our results also provide other experimental researchers with a set of items calibrated for frequency in most Spanish speaking countries.

From a theoretical perspective, these results, together with those from the replication study, would suggest that people speaking a particular language, although in different countries (thus differing in some cultural aspects), share lexical properties of emotion words. Empirical examination of frequency effects show that its effects reflect in part semantic activation, given that lexical access is mediated by the number of contexts in which a word tends to occur rather than pure repetition of occurrence [32]. Thus, these similarities in frequency would tend to agree with the view that emotions constitute basic prototypes [1–4]. Further investigation of empirical semantic judgments in different Spanish speaking countries could evaluate whether there are, in effect, basic semantic similarities, and /or particular nuances in meaning of emotional vocabulary.

## Author Contributions

**Conceptualization:** Ana R. Delgado, Gerardo Prieto, Debora I. Burin.

**Data curation:** Ana R. Delgado, Debora I. Burin.

**Methodology:** Ana R. Delgado, Gerardo Prieto.

**Writing – original draft:** Ana R. Delgado, Debora I. Burin.

**Writing – review & editing:** Ana R. Delgado, Gerardo Prieto, Debora I. Burin.

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
