## [Decision Letter · Decision Letter 0]

23 Jun 2020

PONE-D-20-06126

Agreement on Emotion Labels' Frequency in Eight Spanish Linguistic Areas

PLOS ONE

Dear Dr. Delgado,

First, I would like to apologize again for the delay in the decision. As I mentioned in my earlier correspondence, COVID-19 has disrupted people’s schedule, and to ensure your paper is reviewed by experts in the field, I opted to allow those experts to take longer to review the paper rather than approach less knowledgeable reviewers. I now have the reviews of all three reviewers. As you can see below, the reviewers are highly consistent in the comments that they make. After careful consideration, we feel that it has merit but does not fully meet PLOS ONE’s publication criteria as it currently stands. Therefore, we invite you to submit a revised version of the manuscript that addresses the points raised during the review process. In particular, the two following issues are raised by all and must be addressed before the paper can be published:(1) The paper rests on the assumption that similarity in frequency equals similarity in meanings. All reviewers challenge this assumption. Indeed, its basis is unclear as very different concepts can have similar frequencies, and it is unclear how similarity in frequency can preclude the possibility of a semantic shift. The results therefore cannot support the main argument that the paper makes, namely, that measures such as EVT can be used without adaptation across different Spanish-speaking regions. That said, readers might still find the frequency results of interest, and therefore I would be happy to accept a revised version of the paper that makes a much more constrained claim about similarity in frequency and therefore cross-cultural generalizability regarding familiarity with and knowledge of the terms. Alternatively, you could choose to add other common measures of semantic relatedness (see Reviewer 3’s comments) if you prefer to make the stronger claim and about the suitability of the measure across regions.(2) All reviewers point to the fact that more details are required in order to evaluate the results. In particular, you should provide details about the size of each sub-corpus, the types of texts that each sub-corpus compromises, and the emotion words that were used in the study, including how the words for Study 2 were selected. The reviewers provide many other useful comments that would be good to address. For example, both Reviewer 1 and Reviewer 2 note that your studies don’t align well with the constructionist theory. Therefore, you might want to re-consider the framing and grounding of the studies. If you decide to revise your paper, it would be good to go over all comments and try to address some of them. I hope you find these reviews useful and decide to resubmit a revised version of your manuscript.Bets regards,Shiri Lev-Ari

We look forward to receiving your revised manuscript.

Kind regards,

Shiri Lev-Ari

Academic Editor

PLOS ONE

Journal Requirements:

2. Please improving statistical reporting and refer to p-values as "p<.001" instead of "p=.000". Our statistical reporting guidelines are available at https://journals.plos.org/plosone/s/submission-guidelines#loc-statistical-reporting

Additional Editor Comments (if provided):

Reviewers' comments:

Reviewer's Responses to Questions

**Comments to the Author**

1. Is the manuscript technically sound, and do the data support the conclusions?

Reviewer #1: No

Reviewer #2: Partly

Reviewer #3: Yes

2. Has the statistical analysis been performed appropriately and rigorously? 

Reviewer #1: No

Reviewer #2: Yes

Reviewer #3: No

3. Have the authors made all data underlying the findings in their manuscript fully available?

Reviewer #1: No

Reviewer #2: No

Reviewer #3: Yes

4. Is the manuscript presented in an intelligible fashion and written in standard English?

Reviewer #1: Yes

Reviewer #2: Yes

Reviewer #3: Yes

5. Review Comments to the Author

Reviewer #1: The authors provide ICC analyses describing consistencies in which the frequencies of Spanish emotion words are used across 8 different Spanish speaking regions. They find high intraclass correlations through two different analyses (one with 40 words and one with 60 words, each performed at different dates). The authors conclude that this evidence (1) validates the use of a Spanish emotion vocabulary test across regions and (2) supports the constructionist theory of emotion’s claim that language structures emotion.

The analyses that the authors perform are interesting: Using actuarial linguistic frequency data is not commonly done in psychology, and so this kind of approach is novel and interesting. However, I found the methods and logic of the article very confusing, and I don’t think these analyses support the authors’ conclusions. I explain my argument below with the hopes that these points can help guide the authors’ work.

Major concerns

- I don’t follow the logic that frequency of word usage can be used to validate a vocabulary test. Just because two words are used with similar frequencies doesn’t mean they have the same meanings. To do a thought experiment: Imagine if the word “chair” is used about as often as the word “frustrated.” The authors could then swap the frequencies of the word “chair” out for the word “frustrated” in 4 of the 8 regions and still have exceptionally high ICCs in their analyses. By the authors’ logic, this would mean that the word “chair” is a valid replacement for the meaning of the word “frustrated” in a vocabulary test, but that is clearly not true. I don’t think you can infer meaning of a word from its frequency, meaning that the primary conclusions of the paper do not hold. Now, if the authors want to conduct analyses showing that 60 different emotion words have highly similar frequencies across 8 different regions (and potentially do other follow-up analyses showing which words have most similar vs dissimilar frequencies, etc.) then that could be interesting, but I really don’t think this is a logical approach for testing the validity of a scale across regions.

- I liked how the authors incorporated the Constructionist theory into their work. However, I got lost in understanding the logic of their claim that the Constructionist theory would imply similar emotion word meanings across everyone who speaks Spanish. Yes, language should shape emotion meanings, but 1) similar frequency doesn’t mean similar meaning and 2) the Constructionist theory emphasizes diversity and heterogeneity of emotion word meanings across individuals. Instead of claiming that all people who speak Spanish have identical emotion concepts because they speak Spanish, the Constructionist theory would posit that individuals have highly divergent emotion concepts even within the same region. Additionally, some Social Constructionist scholars would argue that regions should actually differ in how emotion concepts tend to be represented due to cultural factors that differ across regions. I really struggled to follow this part of the authors’ argument, so it seems it should be either argued more clearly or revised.

- The paper is lacking in critical methodological details for us to understand the validity of the authors’ inferences. What kinds of text documents does the corpus draw from (political speeches vs. novels vs. Facebook posts)? What were the 40-60 words used (this must be displayed in the table; coding words by numbers hides essential information from readers)? How exactly did the authors decide which words should be included in the list of 40-60 “emotion” words? What steps were taken to logically think through how word frequencies relate to the authors’ inferences and conclusions? We need more clarity on these points to be able to follow the authors’ methods and logic.

- It doesn't seem reasonable to call the two analyses Study 1 and Study 2. These are two different analyses of the same corpus just using different sets of words across different times. Furthermore, because these analyses draw from the same dataset, it is a little misleading to call the second analysis a “replication” of the first analysis. Instead, I think the authors are reporting 2 analyses within the same study.

Smaller concerns

- It doesn't seem necessary to explain all ICC methods. Just arguing for the approach that is selected is sufficient.

- The abstract is difficult to follow because the methods are not clearly expressed, and the logic connecting the methods to the conclusions is not clear.

- The abstract suggests that the Constructionist theory would favor concordance, but as I argue above, the Constructionist theory emphasizes heterogeneity in emotion concept representation across individuals, languages, and cultures.

- The first paragraph of the intro is a really nice summary of the Constructionist theory, but it unfortunately has lots of jargon and will likely be difficult for non-Constructionist thinkers to follow.

- The paragraph starting at line 73 is hard to follow and could benefit from clearer expression.

- A few other relevant papers to think about are Kalokerinos et al. 2019 in PsychScience (connecting emotion differentiation to regulation); as well as Baron-Cohen et al. 2010 in Frontiers; and Nook, Stavish et al. 2019 in Emotion (studies on emotion vocabulary tests across development).

Reviewer #2: - In the present paper the authors study the normalized frequencies per million in the CORPES XXI corpus of 100 emotion terms in eight Spanish linguistic areas in the world (40 emotion words in the first study and 60 emotion words in the second study). The authors observe an extremely high convergence in this normalized frequencies across these eight areas.

- While the empirical research question and the actual linguistic research are straightforward and generate noteworthy results, the framing and interpretation of the results is problematic.

- In the current paper, the investigation of the normalized frequencies per million of emotion words in the CORPES XXI corpus across the eight linguistic areas is proposed as an alternative to classical adaptation procedures of the stimuli of a psychological assessment instrument. When an assessment instrument will be applied in other cultural and linguistic contexts than for which is was developed, such an adaptation procedure is needed to guarantee the content validity of the instruments for the new contexts in which they will be applied. Moreover, during the adaptation process, in which judgmental evidence is gathered about the stimuli in the instrument, also information is normally collected about the adequacy and appropriateness of the stimuli for the new contexts as well as information about the meaning the stimuli in the other contexts. While the normalized frequencies do give important information about the difficulty of maximum performance items in other cultural and linguistic contexts (the more frequent a word, the more likely a person has learned the meaning of the word and the more easy it is), it is far too strong to claim that this information can be replace the classical adaptation process. The problem is that this frequency does not give any information about possible shifts in meaning of emotion words. For instance, it would not be unlikely that in the US the Spanish “verguënza” is closer to the English word “shame”, that the word “verguënza” in Spain. To justify the use of the same stimuli across cultural and linguistic contexts, it is also very important to study these possible meaning shifts in order to justify their applicability in each of these contexts.

- The current study is framed within a constructivist approach to emotions. However, the results of the current study can be as well, and maybe better, interpreted from a universalist-biological approach to emotions. According to this approach emotions are phylogenetically shaped processes that are engrained in human biological functioning and have been lexically sedimented in language. From this universalist-biological approach strong convergences are expected between cultural and linguistic groups. From a constructivist approach, the meaning of words is expected to be constructed through a process of meaning making which is affected by the surrounding cultural context. The eight Spanish-speaking areas differ substantially in their cultural contexts (as well has the historical developments of their cultural contexts, e.g., exposure to other languages and indigenous cultural groups). One would expect these cultural differences to cause differences in the use of words. The fact that an extremely high convergence is observed in the frequency of the emotion words seems to indicate that these cultural context differences had only limited impact on the use of these words, which does fit more the universalist-biological approach.

- The major weakness of the current paper is that no theoretical framework is presented about what the frequency of use of emotion words mean. Already in the eighties of last centuries (e.g., Fehr & Russell, 1984) it was suggested that the frequency of use of emotion words gave information about the position of emotion words in an hierarchical structure, with more frequent emotion words being more “basic” from a prototype approach [Fehr, B., & Russell, J. A. (1984). Concept of emotion viewed from a prototype perspective. Journal of Experimental Psychology: General, 113, 464-486. doi: 10.1037//0096-3445.113.3.464]. Without a substantial reflection about what the frequency of emotion word use psychologically means, the contribution of the present study is rather limited (emotion words differ in frequency of use, within the Spanish world the frequency of use is very similar, thus items using these emotion words will share the same difficulty across the Spanish world).

- I still have two further points:

o A weakness of both studies is that the authors do not describe how the emotion terms have been selected. Why was it interesting to study these emotion terms? This should be more elaborated, and should be made clear in the text without having to consult third articles.

o The authors do not report the emotion terms themselves, which makes it for the reader an uninteresting paper as she or he cannot judge the content of the results her- or himself. Certainly for the second study there is no good reason not to report the emotion terms as they come from existing published research. Moreover, it is unlikely that by publishing just the emotion words, the actual items of the psychological test are made public. The authors could consider reporting study 1 and 2 together, so that is it not clear which terms stem from the assessment instrument and which terms were studied for other reasons.

Reviewer #3: When a language-based measurement instrument is developed in one language, it typically needs to be adapted for use within other languages, or across use of different cultural groups. This is because direct translations do not necessarily carry the same connotations, and words meanings may shift between different cultural groups, even if they recognizably speak the same language.

The purpose of this research was to quantify the consistency of use, using corpus-based methods, of emotion terms in four different cultural groups that speak spanish. The hypothesis is that if absolute frequencies of use of emotion terms do not vary substantially between groups, then their meanings likely also do not vary between groups, and consequently little or no recalibration would be needed for deploying measurement instruments that use emotion-based language across these groups.

The content and distribution of CORPES XXI is not well-described. The authors should describe the proportional breakdown of their corpus into samples from their eight Spanish subgroups, including absolute number of tokens for each subcorpus. This would be relevant if some subcorpora were small; for instance, Brysbaert and New (2009) have argued that at least 50 million tokens are needed to get an accurate estimate of word frequency.

Certainly, we should expect some words to vary in their frequency of use across different regions/cultures. It is surprising that emotion words are so consistently used. The authors should consider contrasting their results with a similar analysis for non-emotion terms. Or, better yet, the most frequent N (10,000ish) words in each sub-corpus' vocabulary.

The authors should discuss, or rule out, less interesting potential explanations of their results. For instance, if the text in each subcorpus is heavily constrained (e.g., by formalities of speech that span across each subgroup), or if much cross-transfer occurs between the different speaking groups (e.g., if text were from international communication media like social media), these results would be less interesting.

The presented analyses would be strengthened if they were accompanied by a more direct measure of semantic relatedness, e.g., using similarity measures from a distributional semantic model like word2vec, GloVe, or some other model (assuming corpus sizes are sufficient to train such a model). Like the authors say, frequency analyses are only the first step of corpus analyses. Other, fairly easy, steps might also be taken here if the authors have access to enough data.

Finally, analyses like these are generally strengthened when you can demonstrate that they generalize between corpora. I appreciate that the authors may not be able to find additional corpora for all eight subgroups, but if the authors were able to replicate their analyses for even two or three of the subgroups using different corpora, this would greatly strengthen their argument.

6. PLOS authors have the option to publish the peer review history of their article (what does this mean?). If published, this will include your full peer review and any attached files.

Reviewer #1: No

Reviewer #2: No

Reviewer #3: No

---

## [Author Response · Author response to Decision Letter 0]

29 Jul 2020

PONE-D-20-06126

Agreement on Emotion Labels' Frequency in Eight Spanish Linguistic Areas

PLOS ONE

Dear reviewers:

Please, find below a list of modifications (that can be seen highlighted in the Revised manuscript with track changes). Thank you for your suggestions. Most of them have been followed:

(1) In both the Introduction and the Discussion, the interpretation is now constrained to frequency, adding considerations about word frequency effects and relation to semantic features. New references have been added (highlighted in yellow).

(2) Consequently, the framing and grounding of the studies has been reconsidered, and some new references have been added (highlighted in yellow).

(3) Details about the size of each sub-corpus have been provided, as well as the types of texts that each sub-corpus compromises.

(4) We have provided the emotion labels that were used in the two studies, including how the words for Study 2 were selected. 

(5) The data in Table 1 are now presented in alphabetical order instead of the order of entry of the EVT, so that the correct answers to the EVT cannot be known. Please, note that the number of emotion labels in any language is very limited, which is not true of emotion-laden words. One hundred emotion labels that are common enough are not easy to find.

(6) p < .001 is now written instead of p=.00.

(7) The abstract has been rewritten.

We have considered that, in the psychological tradition, analyzing different sets of words gives place to different studies; thus, the structure of the paper has not been changed. We have also considered that many applied researchers can find useful the explanation of all ICC methods. 

I hope that you all find these changes satisfactory enough. 

Best wishes,

---

## [Editor Report · Decision Letter 1]

3 Aug 2020

Agreement on Emotion Labels' Frequency in Eight Spanish Linguistic Areas

PONE-D-20-06126R1

Dear Dr. Delgado,

We’re pleased to inform you that your manuscript has been judged scientifically suitable for publication and will be formally accepted for publication once it meets all outstanding technical requirements.

Kind regards,

Shiri Lev-Ari

Academic Editor

PLOS ONE

Additional Editor Comments (optional):

Dear Dr. Delgado,

Thank you for submitting your revision. I am happy to see that you addressed most of the reviewers' comments including providing additional information, re-situating the study, and revising the main claim and conclusions. I think this version is much better and I am happy to accept it.

Best regards,

Shiri Lev-Ari
---

## [Editor Report · Acceptance letter]

7 Aug 2020

PONE-D-20-06126R1 

Agreement on Emotion Labels' Frequency in Eight Spanish Linguistic Areas 

Dear Dr. Delgado:

I'm pleased to inform you that your manuscript has been deemed suitable for publication in PLOS ONE. Congratulations! Your manuscript is now with our production department. 

Kind regards, 

on behalf of

Dr. Shiri Lev-Ari 

Academic Editor

PLOS ONE